# ReCo: Retrieve and Co-segment
# for Zero-shot Transfer

**Gyungin Shin**[1]
[1]Visual Geometry Group
University of Oxford
UK

**Weidi Xie**[1,2]
[2]Coop. Medianet Innovation Center
Shanghai Jiao Tong University
China

**Samuel Albanie**[3]
[3]Department of Engineering
University of Cambridge
UK

## Abstract

Semantic segmentation has a broad range of applications, but its real-world impact has been significantly limited by the prohibitive annotation costs necessary to enable deployment. Segmentation methods that forgo supervision can side-step these costs, but exhibit the inconvenient requirement to provide labelled examples from the target distribution to assign concept names to predictions. An alternative line of work in language-image pre-training has recently demonstrated the potential to produce models that can both assign names across large vocabularies of concepts and enable zero-shot transfer for classification, but do not demonstrate commensurate segmentation abilities.

We leverage the retrieval abilities of one such language-image pre-trained model, CLIP, to dynamically curate training sets from unlabelled images for arbitrary collections of concept names, and leverage the robust correspondences offered by modern image representations to co-segment entities among the resulting collections. The synthetic segment collections are then employed to construct a segmentation model (without requiring pixel labels) whose knowledge of concepts is inherited from the scalable pre-training process of CLIP. We demonstrate that our approach, termed **Re**trieve and **Co**-segment (ReCo) performs favourably to conventional unsupervised segmentation approaches while inheriting the convenience of nameable predictions and zero-shot transfer. We also demonstrate ReCo's ability to generate specialist segmenters for extremely rare objects.

## 1   Introduction

The objective of *semantic segmentation* is to partition an image into coherent regions and to assign to each region a semantic label. This task has myriad applications across domains such as medical image analysis, autonomous driving, industrial process monitoring and wildlife tracking. However, there are several key challenges that have hindered the deployment of existing semantic segmentation approaches to date: (1) **Cost**: collecting manual pixel-level annotations is extraordinarily expensive (e.g. 90 minutes per image for high quality labels [14]), limiting the use of fully-supervised approaches; (2) **Flexibility**: supervised approaches have typically been trained with limited lists of pre-defined categories and lack the ability to recognise rare or novel categories (such as those described by free-form text); (3) **Complexity of deployment**: unsupervised segmentation methods have dramatically reduced annotation costs, but still exhibit the inconvenience of requiring labelled examples to assign names to predictions; (4) **Data access**: many existing approaches (both supervised and unsupervised) are trained on the *target data distribution*, requiring both that this distribution is known at training time (limiting flexibility) and that this data is accessible, which may not be the case for legal/ethical reasons (e.g. medical image data).

There is a rich body of semantic segmentation literature that proposes solutions to subsets of these challenges, but to our knowledge no existing work addresses their full combination. To tackle all four challenges, we draw inspiration from two lines of recent research. The first line of research has

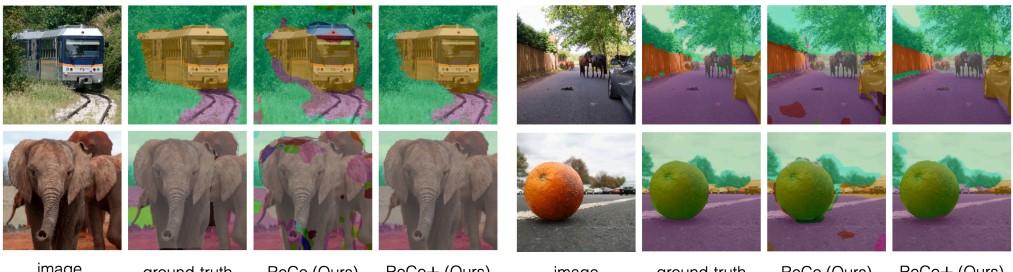

| image | ground-truth | ReCo (Ours) | ReCo+ (Ours) | image | ground-truth | ReCo (Ours) | ReCo+ (Ours) |

Figure 1: We propose ReCo, a new framework for semantic segmentation zero-shot transfer without pixel supervision. The figure depicts ReCo segmentations for COCO-Stuff [6], indicating promising results in the challenging zero-shot transfer setting. We also illustrate results with unsupervised adaptation to the target distribution (ReCo+), trading flexibility for improved segmentation quality.

shown that modern deep architectures (most notably vision transformers [18]) develop the ability to infer the spatial extent of objects without pixel supervision [7, 60, 79, 86, 77, 55] and moreover can establish semantically consistent correspondences across images [13, 24]. The second line of research has demonstrated that large-scale visual-language pre-training [71, 35] produces models that possess both a large vocabulary and remarkable *zero-shot transfer* potential—the ability to recognise concepts on target datasets without access to the target data distribution during training.

In this work, we target a synthesis of these two approaches that draws on their respective strengths for the semantic segmentation task. We first employ CLIP [71] to curate training sets from unlabelled images for any desired list of concepts. We then exploit the robust semantic correspondences offered by modern vision backbones to co-segment concepts among the resulting curated collections. Finally, we use these co-segmented concepts to construct a segmentation model for the given concepts, *without* requiring training. This framework, which we term **Re**trieve and **Co**-segment (ReCo), performs favourably to existing unsupervised segmentation approaches while preserving the benefits of a wide vocabulary, named predictions and zero-shot transfer exemplified by CLIP. Since ReCo utilises a pre-trained image-language model, we call the task considered in this work *unsupervised semantic segmentation with language-image pre-training* to differentiate it from conventional unsupervised semantic segmentation task formulations.

Our contributions are three-fold: (1) We propose the ReCo framework, enabling open vocabulary semantic segmentation, without pixel annotations or the need to provide labelled examples from the target domain thereby enabling zero-shot transfer; (2) We compare our approach to prior work on standard semantic segmentation benchmarks (COCO-Stuff [6], Cityscapes [14], and KITTI-STEP [87]), and further illustrate the ability of ReCo to segment rare concepts beyond these benchmarks; (3) For cases when the target image distribution *is* available, we demonstrate that a simple extension of our approach, ReCo+, can exploit this data access via unsupervised adaptation to bring further gains.

## 2 Related work

Our work is connected to several themes in the literature, which we describe next.

**Unsupervised semantic segmentation.** There has been considerable recent progress towards unsupervised semantic segmentation with deep neural networks by leveraging ideas from self-supervised learning. Learning objectives based on maximising mutual information between views [34, 61], metric learning across proposals [93, 82], equivariance and invariance constraints [12], distillation of self-supervised feature correspondences [24] and cross modal cues (vision and LiDAR) [85] have all shown their potential for this task.

One drawback of these approaches is a reliance on either nearest neighbour search on a held-out set with pixel level annotations or the Hungarian algorithm [45] (optimally matching predictions against ground-truth semantic masks) to produce segments with names. In contrast, by leveraging the ability of a language-image model to name concepts, ReCo is independent of labelled examples during both training and inference.

**Weakly-supervised semantic segmentation.** To reduce annotation costs, a number of works have explored weaker cues such eye tracking [63], pointing [4, 70, 11], sparse pixel labels [78], scribbles [49], web-queried samples [36], boxes [15, 42, 80], extreme clicks [64, 54], image-level labels [95, 88, 19, 1, 8, 67, 69] and free-form text [90]. However, such approaches still require

the weak annotation to be attached to the data used to train the segmentation model—by contrast, ReCo can in principle train a segmenter from any unlabelled collection of images. ReCo also bears a conceptual similarity with *webly-supervised* approaches [20] for semantic segmentation [36, 76]. These methods employ an image search engine such as Google to provide training samples for concepts. However, their flexibility is limited (this approach cannot be applied to private or commercially sensitive data, for example) and they lack the ability to leverage the knowledge of the search engine itself during inference (we demonstrate that integrating the vision-language model into the inference procedure brings significant gains in performance).

**Zero-shot semantic segmentation with pretrained language/vision-language embeddings.** A diverse body of work has explored zero-shot semantic segmentation, broadly defined as the task of segmenting categories for which no labels were provided during training (often termed *zero-label* semantic segmentation [89]). The key idea underlying many of these works is to leverage relationships encoded in pretrained word embeddings (such as word2vec [56] or GloVe [68]) to enable generalisation to unseen categories [94, 5, 23, 89, 40, 29, 47, 65]. More recent work has sought to leverage the vision-language embedding space learned by CLIP [71] to improve dense prediction in various settings [96, 72, 46, 91, 17, 52]. We adopt a variation of DenseCLIP [96] as a component of our framework. Additionally, differently from the above, we pursue the formulation of *zero-shot transfer* popularised by CLIP [71] which evaluates performance on *unseen datasets* rather than *unseen categories*. Consequently, unlike these works, our model has no access to either labelled or unlabelled examples from the target data distribution (or pixel-level labels from the source distribution, as investigated by [91]). We note one exception: in addition to their primary zero-shot evaluations, DenseCLIP [96] also report an "annotation-free" evaluation without access to the target dataset—we compare our approach with theirs under an equivalent setting.

**Large-vocabulary/rare concept segmenters.** To scale up the number of concepts that can be segmented by a model several strategies based on captions [21], grounded text descriptions [39] and annotation transfer [30, 32] have been explored. In a differing direction, various losses and incremental learning techniques have been employed [31, 27] to better segment rare concepts. Unlike ReCo, however, each of the above approaches still requires costly pixel-level annotations.

**Co-segmentation** which aims to segment common regions among a collection of images, has been widely studied with classical computer vision approaches [73, 59, 83, 3, 38, 84]. The topic has been revisited with deep learning using shared encoder networks [9, 48, 2, 51], iterative refinement [92] and weak (class-label) supervision [28]. While ReCo can in principle make use of any unsupervised co-segmentation algorithm, we find that a simple correlation strategy works well, and thus we adopt it for our approach.

## 3   Method

In this section, we first formalise the task of unsupervised semantic segmentation with language-image pre-training (Sec. 3.1). We then introduce the ReCo framework (Sec. 3.2) which enables zero-shot transfer for semantic segmentation with arbitrary categories. We describe a language-based gating mechanism to enhance segmentation quality (Sec. 3.3), and a pseudo-labelling scheme, ReCo+, which adapts to a target distribution using predictions from ReCo (Sec. 3.4).

### 3.1   Unsupervised semantic segmentation with language-image pre-training

Let us denote by $x \in \mathbb{R}^{3 \times h \times w}$ an image of interest. Let $\Omega = \{1, \ldots, h-1\} \times \{1, \ldots, w-1\}$ denote its spatial domain and $\omega \in \Omega$ a pixel location. The objective of *unsupervised semantic segmentation with language-image pre-training* (USLIP) is to assign to each pixel location $\omega \in \Omega$ a category, $c \in C$, that falls among one of $|C|$ mutually exclusive target categories. The key characteristic distinguishing USLIP from traditional unsupervised segmentation is that in order to tackle the task, a model may assume access to *pre-training corpus* of paired image and text data, without *pixel-level supervision*. We distinguish this approach from weakly-supervised learning in that it does not rely on weak human annotations (*e.g.,* image-level labels) during training for a downstream task (*i.e.,* segmentation).

### 3.2   Retrieve and Co-segment (ReCo)

The inputs to ReCo are a collection of unlabelled images, and a list of text descriptions of concepts to be segmented. Through a combination of *image retrieval* and *co-segmentation* across the image

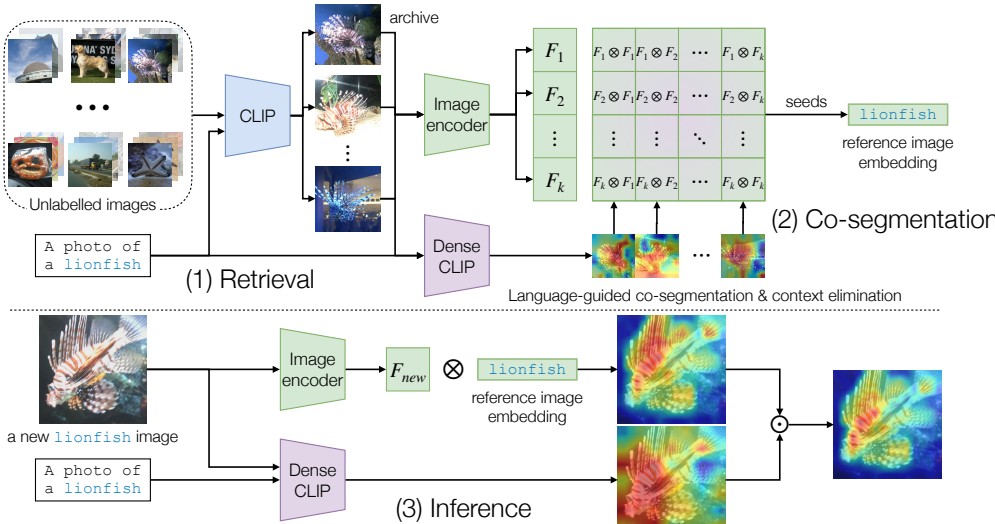

Figure 2: In this work, we propose ReCo, a framework for open vocabulary semantic segmentation zero-shot transfer utilising a language-image pre-training model (*i.e.,* CLIP). **Top:** (1) Given a large-scale unlabelled dataset and a category to segment, we first curate an archive of $k$ images from the unlabelled collection using CLIP [71]. (2) Using a pre-trained visual encoder (*e.g.,* MoCov2 [25], DeiT-S/16-SIN [60]), we extract dense features from the archive images, which are used to generate a reference image embedding for the given category via a co-segmentation process. **Bottom:** (3) During inference, the reference image embedding is employed to produce an initial segmentation of the target concept which is refined with DenseCLIP. $\otimes$ and $\odot$ denote inner product and Hadamard product, respectively. See Sec. 3 for details.

collection, ReCo constructs a segmenter for the given concepts on the fly. During *inference*, this segmenter is applied *without* fine-tuning to images from a target distribution of interest, thus supporting the zero-shot transfer. The interaction of these three stages is illustrated in Fig. 2 and discussed in more detail below.

**Curating exemplars through image retrieval.** The first hypothesis underpinning our approach is that it is possible to construct a sufficiently large and diverse unlabelled image collection that contains examples (or closely related examples) of any concept we wish to segment. We consider this hypothesis to be reasonable in light of the fact that recent research has produced a number of diverse datasets spanning billions of images [53, 22, 35, 75].

To make use of this hypothesis for segmentation, we employ a vision-language model to curate an archive of images for each concept of interest. In this work, we use CLIP [71], a model comprising an image encoder and a text encoder that enables efficient image retrieval for free-form text queries. In more detail, for a given text query describing a concept $c$ the CLIP text encoder $\psi_{\mathcal{T}}$ produces an embedding $\psi_{\mathcal{T}}(c) \in \mathbb{R}^e$ which can be compared against the embeddings produced by the image encoder $\psi_{\mathcal{I}}$ for each image $x$ in the unlabelled collection $\mathcal{U}$: $\{\psi_{\mathcal{I}}(x) \in \mathbb{R}^e : x \in \mathcal{U}\}$. We then build an archive from the $k$ images that form nearest neighbours for each concept $c$. While a more sophisticated strategy (for instance, adjusting the archive size according to concept difficulty or prevalence) is possible with ReCo, we found that this simple approach worked well, and so we adopt it here. Note that, thanks to mature approximate nearest neighbour search techniques [37], this archive construction process can be readily applied to a collection $\mathcal{U}$ containing billions of images.

**Co-segmentation with seed pixels.** The second hypothesis underpinning our approach is that modern vision-language models such as CLIP are capable of constructing archives of *high purity* (i.e. a high ratio of the archive images contain the concept of interest) provided that such exemplars exist in the underlying image collection $\mathcal{U}$ (our first hypothesis). Our second hypothesis, which we validate through experiments in Sec. 4, enables the use of *co-segmentation* to obtain concept segments.

Concretely, given the $k$ images comprising the archive for category $c$ constructed by CLIP, we aim to segment regions corresponding to concepts that re-occur across the images. Since we expect the archive to be of high purity, and since modern visual backbones provide consistent semantic correspondences across images [24], we adopt a strategy of first identifying a *seed pixel* in each

image that we are confident belongs to the target category, $c$. Intuitively, good seed pixels are ones that have close neighbours (i.e. strong support) across each of the archive images, since we assume that: (i) the target concept is common to all images in the archive, (ii) our visual backbone will produce consistent features for pixels belonging to the same concept. We then use these seed pixels to construct a reference embedding that can be used to classify pixels belonging to $c$.

In detail, we first extract dense features for each image in the archive using a pre-trained image encoder $\phi_{\mathcal{I}}$:

$$\{F_1, \ldots, F_k\} = \{\phi_{\mathcal{I}}(x_1), \ldots, \phi_{\mathcal{I}}(x_k)\} \tag{1}$$

where $F_i \in \mathbb{R}^{d \times h \times w}$ denotes (spatially) dense features for the $i^{th}$ image with height $h$, width $w$ and $d$ channels. Each such feature is L2-normalised along its channel dimension.[1] Note that any image encoder $\phi_{\mathcal{I}}$ can be employed here (it need not be CLIP).

We identify seed pixels in four steps: First, we construct an adjacency matrix $A^{khw \times khw}$ among all features in the archive. Here, each of the $k \times k$ submatrices, $A_{ij} \in \mathbb{R}^{hw \times hw}$, encodes pairwise similarities between features from image $i$ and image $j$ for $i, j \in \{1, ..., k\}$. Second, we aim to identify, for each pixel in the archive, the similarity of its nearest neighbour among each of the $k$ images. To do so, we apply a max operator along the columns of each submatrix (reducing the $hw$ colums of each submatrix to 1 and reducing the overall adjacency matrix dimensions to $khw \times k$). Third, we aim to identify the average *support* that each pixel has among the $k$ archive images. For this, we apply a mean operator over the columns of each submatrix such that each row of the resulting $khw \times 1$ matrix encodes the mean maximum similarity across $k$ images. Finally, we identify the seed pixel locations by applying an argmax operator to each of the $k$ submatrices of size $hw \times 1$, yielding the spatial indices of the features in each image with highest average maximum similarity across the archive.

To construct a classifier for concept $c$, we simply average the embeddings of the $k$ seed pixels from its archive and L2-normalise the resultant vector to produce the reference embedding $f_c \in \mathbb{R}^d$.

**Inference.** To localise instances of the category in a new image $x_{\text{new}}$, we first compute the dot-product between $f_c$ and the L2-normalised dense features $F_{\text{new}}$ from the new image, and pass the result through a sigmoid:

$$P_{\text{new}}^c = \sigma(f_c \cdot F_{\text{new}}) \in [0, 1]^{h \times w} \tag{2}$$

where $P_{\text{new}}^c$ denotes an initial estimate of the probability map corresponding to category $c$.

To refine this probability map, we draw inspiration from recent work [96, 72] showing that dense visual CLIP features can be usefully correlated against a given CLIP text embedding. Concretely, we employ the DenseCLIP mechanism of Zhou et al. [96] to highlight regions of the input that are salient for the target category $c$ as follows. For image $x_{\text{new}}$, we extract features $V_{\text{new}} \in \mathbb{R}^{e_v \times h \times w}$ from the last self-attention layer values of the CLIP image encoder (here $e_v$ denotes the value feature dimension), project the features into the joint space $\mathbb{R}^e$ with the CLIP image encoder's final linear projection and L2-normalise the result. We then compute the CLIP text embedding $\psi_{\mathcal{T}}(c) \in \mathbb{R}^e$ for the target concept $c$ and L2-normalise it before producing a saliency map $\mathcal{S}_{\text{new}}^c \in [0, 1]^{h \times w}$ by applying the text embedding as a $1 \times 1$ convolution to the visual features and applying a sigmoid activation function to the result. Note that our use of a sigmoid activation function differs from the softmax used by [96], since we process each concept independently.

Our final probability map for category $c$ is produced by the Hadamard product of these estimates:

$$\bar{P}_{\text{new}}^c = P_{\text{new}}^c \odot \mathcal{S}_{\text{new}}^c \tag{3}$$

In case of multiple categories predictions, we concatenate all the category prediction maps and apply argmax to the category dimension. As a simple post-processing step, we also experiment with the effect of applying a CRF [44] (similarly to [24]). Pseudocode for ReCo can be found in the supplementary material.

### 3.3 Language-guided co-segmentation and context elimination

Even when an archive has high purity, co-segmenting the target concept $c$ from the collection of images can be challenging due to the potential presence of distractor categories $\tilde{c}$ that often co-occur

---

[1]For clarity purposes, we assume that all images share the same spatial extent (and thus their dense features also do). However, the proposed co-segmentation approach can be applied to images with different resolutions.

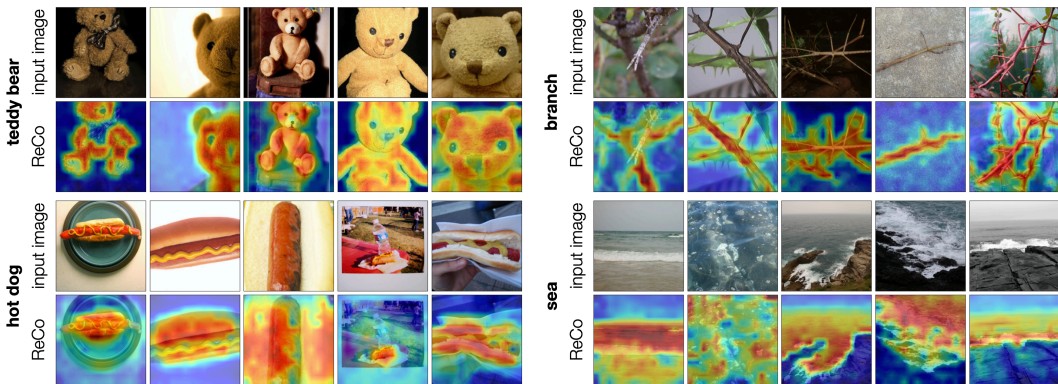

Figure 3: **Sample visualisations of the co-segmentation approach used by ReCo.** We display the top 5 ranked images among an archive of 50 retrieved images for a given concept. Highlighted regions are shown in red. For both *things* (left) and *stuff* (right) categories, ReCo reliably recognises appropriate regions. Best viewed in colour.

with $c$. For example, *cars* often co-occur with *roads* and likewise *aeroplanes* with *sky*. To minimise the risk of the co-segmentation algorithm anchoring on unintended categories, we introduce two mechanisms into the co-segmentation procedure, namely, *language-guided co-segmentation* and *context elimination*.

**Language-guided co-segmentation.** Specifically, for each $x_i$ in the archive, we first compute the corresponding saliency map $\mathcal{S}_i^c \in [0,1]^{h \times w}$ using DenseCLIP. We then vectorise this map and use it to filter the co-segmentation similarities by replacing each row of $A_{j,i}$ with the Hadamard product of itself with $\text{vec}(\mathcal{S}_i^c) \in [0,1]^{1 \times hw}$ (where $\text{vec}(\cdot)$ denotes vectorisation) for each submatrix $j = \{1, ..., k\}$. Intuitively, this serves to act as a *gate* that preserves similarities only for pixels identified by CLIP as salient for the target concept.

**Context elimination.** To reduce the influence of widely co-occurring distractors, we select a few common background categories $\tilde{c}$ that frequently appear in images (*e.g., tree, sky, road*) and compute their reference embeddings $f_{\tilde{c}}$ as described in the seeding procedure of Sec. 3.2. We then use the resulting attention maps $P_i^{\tilde{c}}$ produced by these reference embeddings to suppress regions of common background context in the subsequent co-segmentation processes for different categories. This is done similarly as above, by vectorising and replacing each row of $A_{j,i}$ with the Hadamard product of itself with $\text{vec}(1 - P_i^{\tilde{c}}) \in [0,1]^{1 \times hw}$ for each submatrix $j = \{1, ..., k\}$. If a target concept $c$ is identical to one of the common background categories $\tilde{c}$, we replace $f_c$ with the corresponding $f_{\tilde{c}}$.

### 3.4 ReCo+: Fine-tuning ReCo with pseudo-labels

The ReCo framework outlined above requires no training and no access to the target distribution. However, it is possible to consider a simple extension to this idea when access to the target image distribution is available. Our proposed extension, ReCo+, simply trains a segmentation architecture (*e.g.,* DeepLabv3+ [10]) with the segmentation masks from ReCo as pseudo-masks.

## 4 Experiments

In this section, we start by describing the datasets used for our experiments (Sec. 4.1) and implementation details (Sec. 4.2). Then, we conduct an ablation study (Sec. 4.3) and compare our model to state-of-the-art methods for unsupervised semantic segmentation with and without language-image pre-training (Sec. 4.4). Finally, we showcase our model's ability to segment rare-category objects (Sec. 4.5).

### 4.1 Datasets

For our ablation study, we use the ImageNet1K [16] validation set to curate archive for concepts of interest. The dataset covers 1K classes with 50 images for each class. To measure segmentation performance in the zero-shot transfer setting, we use the PASCAL-Context [58] validation set for evaluation, which has 5,104 images of 59 categories excluding the background class.

To compare with previous unsupervised segmentation methods, we use the ImageNet1K training set to construct the reference embedding for the concept we wish to segment. The dataset has 1K classes with 1.2M images. We evaluate on standard benchmarks including the Cityscapes [14] validation

split, which has 500 urban scene images of 27 categories, KITTI-STEP [87] validation set, which is composed of 2,981 urban scene images of 19 categories, and COCO-Stuff [6] validation split, which has 4,172 images of 171 low-level thing and stuff categories excluding background class. Following [12, 24], we use the 27 mid-level categories for evaluation. For unsupervised adaptation with ReCo+ (Sec. 3.2), we train on ReCo pseudo-labels on the Cityscapes training set with 2,975 images, KITTI-STEP training set which contains 5,027 images, and the COCO-Stuff10K subset which has 9,000 images for each respective benchmark. We emphasise that *no ground-truth labels are used for training.*

Finally, to demonstrate our model's ability to segment rare concepts, we use the LAION-5B dataset [75] with 5 billion images as a large collection of images that we expect to satisfy our first hypothesis, namely that it will have coverage of rare concepts. To assess performance, we use the FireNet dataset [62] which has 1,452 images spanning rare fire safety-related classes. For our experiment, we select the *fire extinguisher* class as an example of a concept that is important but rare in vision datasets (it is not contained in ImageNet1K [74], for example) and evaluate ReCo on 263 images containing at least one instance of the category. As a further proof of concept, we also demonstrate co-segmentations of the *Antikythera mechanism* (a historical item that does not appear in WordNet [57], or any labelled vision datasets that we are aware of).

## 4.2 Implementation details

Here, we describe the hyperparameters used to train ReCo+, inference details and evaluation metrics. Our implementation is based on the PyTorch library [66] and made publicly available.[2]

**ReCo+ Training.** While ReCo does not require training, we train ReCo+ based on the DeepLabv3+ [10] segmentation architecture with a ResNet101 [26] backbone on the predictions from ReCo as described in 3.2. All training images are resized and center-cropped to $320{\times}320$ pixels and data augmentations such as random scaling, cropping, and horizontal flipping are applied with random color jittering and Gaussian blurring. We use the Adam optimiser [43] with an initial learning rate of $5 \times 10^{-4}$ and a weight decay of $2 \times 10^{-4}$ with the Poly learning rate schedule as in [50, 10]. Training consists of 20K gradient iterations with a batch size of 8 and takes about 5 hours on a single 24GB NVIDIA P40 GPU.

**Inference.** For each benchmark, we pre-compute reference image embeddings for a list of categories for the benchmark and store the embeddings to form a classifier. Whenever DenseCLIP is employed, we use the ResNet50x16 model (following [96]) to construct a saliency map for each image. Unless otherwise stated, for the COCO-Stuff and Cityscapes benchmarks, we resize and center crop the input images to $320{\times}320$ pixels as in [24]. For the KITTI-STEP validation set, we use the original resolution of each image as in [41]. For the FireNet benchmark, we resize the shorter side of images to 512 pixels and predict a single class of *fire extinguisher* by thresholding the predicted heatmap with probability of 0.5.

**Evaluation metrics.** Following the common practice [34, 12, 24], we report pixel accuracy (Acc.) and mean intersection-over-union (mIoU).

## 4.3 Ablation studies

**Ability of CLIP to curate archives.** We begin by assessing the validity of our second hypothesis—namely that CLIP is capable of achieving *high purity* archives from unlabelled images. To this end, we evaluate the retrieval performance of different CLIP models on the ImageNet1K validation set when constructing different archive sizes. In detail, for each archive size, $k$, we compute the precision of the top-$k$ retrieved images based on whether the the ground-truth image-labels match the query text. As can be seen in Fig. 4 (left), all CLIP models achieve solid retrieval performance, suggesting their potential for curating high purity archives as part of ReCo. Since ViT-L/14@336px performs best, we employ this as our retrieval model in the remaining experiments.

**Influence of archive size and visual encoder used for co-segmentation.** In Fig. 4 (right) we illustrate the effect of using different pre-trained architectures, *e.g.* MoCov2 [25], DINO [7], CLIP [71], DeiT-SIN [60], as the archive size (and thus the number of images used for co-segmentation) changes. The y-axis depicts segmentation performance for ReCo with these configurations on the PASCAL-Context benchmark. We observe that using larger archives tends to improve performance (likely due to their reasonably high purity) albeit non-monotonically, and that features from DeiT-SIN perform best. We

---

[2]Code available at https://github.com/NoelShin/reco

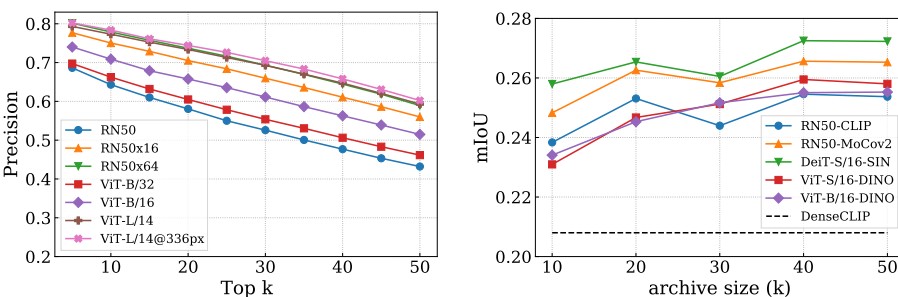

Figure 4: **Ablation studies. Left:** Image retrieval performance of different CLIP models on the ImageNet1K validation set with $k$ ranging from 5 to 50. ViT-L/14@336px performs particularly strongly, suggesting the ability to curate archives of high purity. **Right:** Co-segmentation performance on PASCAL-Context validation set as we vary the archive size and choice of visual encoders. We observe a general trend towards improved performance with increasing archive size for all encoders.

| DenseCLIP | LGC | CE | CRF | Acc. | mIoU |
|:---:|:---:|:---:|:---:|:---:|:---:|
| ✗ | ✗ | ✗ | ✗ | 16.8 | 5.7 |
| ✓ | ✗ | ✗ | ✗ | 41.1 | 21.8 |
| ✓ | ✓ | ✗ | ✗ | 43.1 | 23.1 |
| ✓ | ✗ | ✓ | ✗ | 49.7 | 26.0 |
| ✓ | ✓ | ✓ | ✗ | 50.9 | 26.6 |
| ✓ | ✓ | ✓ | ✓ | **51.6** | **27.2** |

Table 1: **Influence of ReCo components for zero-shot transfer on PASCAL-Context [58].** We observe that integrating DenseCLIP during inference, *Language-guided co-segmentation* (LGC), *Context elimination* (CE), and CRF [44] post-processing each contribute to improved performance. All comparisons use a DeiT-SIN visual backbone for co-segmentation and ViT-L/14@336px for archive curation.

therefore use DeiT-SIN with an archive size of $k$=50 for the remaining experiments unless otherwise stated. See Fig. 3 for qualitative examples of our co-segmentation results.

**Influence of ReCo framework components.** We next assess the effect of employing DenseCLIP during inference, the *language-guided co-segmentation* and *context elimination* components of ReCo which seek to improve the quality of co-segmentation achieved across each archive to boost downstream segmentation performance. When applying context elimination, we select *tree, sky, building, road*, and *person* as common background concepts appearing in natural images to be suppressed. In Tab. 1, we show the effect of three of these strategies, together with the effect of applying a CRF [44] as post-processing. We observe that integrating DenseCLIP into the inference procedure brings a significant gain in performance which we believe is driven by the notable robustness of CLIP features under zero-shot transfer [71]. In addition, language-guided co-segmentation and context elimination further boost co-segmentation performance, while the CRF brings a small gain. We therefore use each of these strategies (including CRF post-processing) in the remaining experiments.

### 4.4 Comparison to state-of-the-art unsupervised methods

We compare ReCo and ReCo+ to state-of-the-art unsupervised semantic segmentation models with and without vision-language pre-training on standard benchmarks, including COCO-Stuff [6], Cityscapes [14] and KITTI-STEP [87] under both zero-shot transfer and unsupervised adaptation (training without labels on the target distribution). For COCO-Stuff, we observe that the mid-level categories used for evaluation are somewhat abstract for retrieval (for instance, one mid-level category is "outdoor objects", which may include many low-level categories beyond the target hierarchy). To avoid introducing ambiguity to the co-segmentation procedure, we instead directly use the low-level categories and then merge the predictions into the mid-level categories. Additionally, we rephrase two category names to reduce ambiguity (*parking* to *parking lot* and *vegetation* to *tree*) in Cityscapes and KITTI-STEP based on the descriptions found in [14]. A detailed discussion can be found in the supplementary material.

As shown in Tab. 2, ReCo strongly outperforms prior models on all benchmarks for zero-shot transfer. Under an unsupervised adaptation protocol, ReCo+ outperforms the state-of-the-art by a large margin on the Cityscapes and KITTI-STEP. On COCO-Stuff, ReCo+ achieves slightly lower pixel accuracy

| Model | Acc. | mIoU |
|---|---|---|
| **Zero-shot transfer** | | |
| DenseCLIP‡ [96] | 32.3 | 19.8 |
| ReCo‡ (Ours) | **46.6** | **27.2** |
| **Unsupervised adaptation** | | |
| IIC [34] | 21.8 | 6.7 |
| MDC [12] | 32.2 | 9.8 |
| PiCIE [12] | 48.1 | 13.8 |
| PiCIE + H [12] | 50.0 | 14.4 |
| STEGO [24] | **56.9** | 28.2 |
| ReCo+‡ (Ours) | 54.5 | **33.0** |

| Model | Acc. | mIoU |
|---|---|---|
| **Zero-shot transfer** | | |
| DenseCLIP*‡ [96] | 35.9 | 10.0 |
| MDC*† [12] | - | 7.0 |
| PiCIE*† [12] | - | 9.7 |
| D&S*† [85] | - | 16.2 |
| ReCo*‡ (Ours) | **65.4** | **22.0** |
| **Unsupervised adaptation** | | |
| IIC [34] | 47.9 | 6.4 |
| MDC [12] | 40.7 | 7.1 |
| PiCIE [12] | 65.5 | 12.3 |
| STEGO [24] | 73.2 | 21.0 |
| ReCo+‡ (Ours) | **83.7** | **24.2** |

| Model | Acc. | mIoU |
|---|---|---|
| **Zero-shot transfer** | | |
| DenseCLIP‡ [96] | 34.1 | 15.3 |
| ReCo‡ (Ours) | **70.6** | **29.8** |
| **Unsupervised adaptation** | | |
| SegSort [33] | 69.8 | 19.2 |
| HSG [41] | 73.8 | 21.7 |
| ReCo+‡ (Ours) | **75.3** | **31.9** |

Table 2: **Comparison to state-of-the art approaches on COCO-Stuff (left), Cityscapes (middle), and KITTI-STEP (right) validation sets.** *Evaluated at the original resolution. †Models trained on Waymo Open [81] (reported from [85]). ‡Models that leverage a language-image pre-training model to assign a concept name to a prediction. The best score for each metric under each protocol is highlighted in **bold**. We observe that ReCo and ReCo+ perform strongly relative to prior work under zero-shot transfer and unsupervised adaptation protocols, respectively.

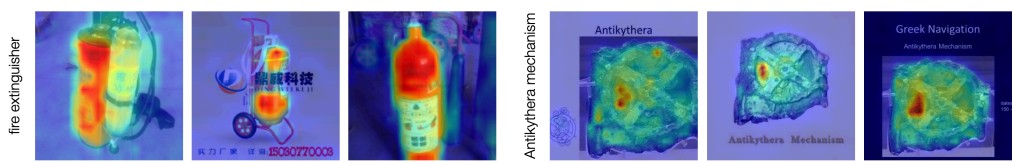

Figure 5: **Co-segmentations of rare concepts. Left:** *fire extinguisher*. **Right:** *Antikythera mechanism*. We show selected samples from ReCo archives for each concept, together with their co-segmentations. In each case, ReCo successfully identifies the regions associated with the concept.

compared to [24] but a considerably higher mIoU. In Fig. 1, we visualise the sample predictions of our models on COCO-Stuff.

### 4.5 Segmenting rare concepts

By virtue of inheriting CLIP's diverse knowledge of nameable visual concepts, ReCo exhibits the ability to segment rare categories. We first demonstrate this ability for *fire extinguisher* objects, which have important fire-safety implications but seldom appear in popular semantic segmentation benchmarks. To assess performance, we evaluate segmentation quality on the FireNet dataset (as described in Sec. 4.1) and achieve reasonable performance on pixel accuracy (93.3) and IoU (44.9) metrics. In Fig. 5 (left) we visualise the co-segmentation produced by ReCo across sample images container fire extinguishers. As an additional demonstration (Fig. 5, right), we also show co-segmentations for images containing the views of the (unique) *Antikythera mechanism*. In both cases, we observe that ReCo is capable of co-segmenting the concept of interest without labelled examples.

## 5 Limitations

Our work has several limitations: (1) There may be cases for which our first hypothesis does not hold—concepts so rare that they do not appear in billion-image scale datasets. In such cases, *e.g.* "a purple elephant with square orange feet wearing an inverted cowboy hat in front of the Doge's Palace", ReCo will struggle. (2) During inference, we make use of a visual encoder and a pre-trained language-image model, *i.e.,* CLIP (a computationally heavy model). Future work could address this with training a light student network which distills knowledge from both models. (3) We employ ImageNet as the unlabelled image collection for our primary investigations, which is known to exhibit an object-centric bias. While qualitative results suggest that ReCo can learn from extremely diverse data [75], a more comprehensive empirical evaluation would strengthen this result. (4) Models like CLIP are expensive to train and are consequently updated infrequently. If a new concept emerges (e.g. a new product) after CLIP was trained, ReCo will be unable to build an archive to enable co-segmentation for this concept. (5) Although our work is unsupervised in the sense that it is

not trained on pixel-level annotation, the ablation studies on PASCAL-Context have guided the development of our method, and thus it benefits from a form of indirect supervision. As a result, our design likely contains choices that subtly favour performance on the evaluation tasks.

## 6    Broader impact

Semantic segmentation is a *dual use* technology. It enables many applications with the potential for significant positive societal impact across domains in medical imaging, wildlife monitoring, improved fault detection in manufacturing processes etc. However, it is also vulnerable to abuse: it may enable unlawful surveillance or invasions of privacy, for example. By removing the requirement to collect pixel masks for concepts of interests, ReCo lowers the barrier to entry for any individual that wish to make practical use of segmentation, but makes no distinction on the ethical implications of the use case, positive or negative.

ReCo makes use of large-scale, unlabelled image collections. By their nature, such images are subject to minimal curation and sanitisation, and thus may contain not only biases across demographics, but also content that does not align with the ethical values of the user. Consequently, we emphasise that ReCo represents a research proof of concept that is not appropriate for real-world usage without extensive additional analysis of the specific deployment context in which it will be used and, in particular, safeguards to moderate the archive curation process.

## 7    Conclusion

In this work, we introduced the ReCo framework for semantic segmentation zero-shot transfer. By drawing on the strengths of large-scale language-image pre-training and modern visual backbones, ReCo attains the ability to segment rare concepts and to directly assign names to concepts without labelled examples from the target distribution. In addition, experimental comparisons demonstrate that ReCo strongly outperforms existing zero-shot transfer approaches that forgo pixel supervision.

**Acknowledgements and disclosure of funding.** This work was performed using resources provided by the Cambridge Service for Data Driven Discovery (CSD3) operated by the University of Cambridge Research Computing Service (www.csd3.cam.ac.uk), provided by Dell EMC and Intel using Tier-2 funding from the Engineering and Physical Sciences Research Council (capital grant EP/T022159/1), and DiRAC funding from the Science and Technology Facilities Council (www.dirac.ac.uk). GS is supported by AI Factory, Inc. in Korea. GS would like to thank Yash Bhalgat, Tengda Han, Charig Yang, Guanqi Zhan, and Chuhan Zhang for proof-reading. SA would like to acknowledge the support of Z. Novak and N. Novak in enabling his contribution.

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
