# ReCo: Retrieve and Co-segment
# for Zero-shot Transfer
# Supplementary Material

**Gyungin Shin**[1]
[1]Visual Geometry Group
University of Oxford
UK

**Weidi Xie**[1,2]
[2]Coop. Medianet Innovation Center
Shanghai Jiao Tong University
China

**Samuel Albanie**[3]
[3]Department of Engineering
University of Cambridge
UK

## Contents

In this supplementary material, we first discuss the role of supervisory signals for ReCo and alternative approaches, and the datasets used in our work (Appendix A). Next we provide further details of experiments conducted in the main paper together with hyperparameters for training ReCo+ in Appendix B. We then report additional ablations investigating the influence of common category selection for context elimination, reducing ambiguity in category name, architectural chioces for CLIP and DenseCLIP, and the number of seed pixels considered for computing a reference image embedding in Appendix C. Finally, we provide additional qualitative results in Sec. D to illustrate both successful and failure cases for our method.

# A    Discussion of supervision and data

## A.1    Supervisory signals for ReCo and prior work

In the main paper, we compare to previous methods that are typically described as *unsupervised.* In practice, however, many methods (including ours) either implicitly or explicitly engage humans in the data curation process at some stage.

*Supervision used by ReCo.* (1) Similarly to prior work, our experiments make use of datasets constructed from photographs taken by humans for both training and evaluation. These photographs are naturally biased towards content that humans find interesting and are typically well-framed (with a concept of interest featuring prominently) or taken from a vantage point that offers a convenient scene overview (e.g. a roof-mounted camera on a vehicle driving on public roads). By training and evaluating on such data, our experimental results likely provide an optimistic assessment of performance when contrasted with other distributions (e.g. the video feed received by an autonomous mobile robot). (2) For our comparisons to prior work, we use the ImageNet training subset without labels to curate archives. However, in practice, this dataset is not free from human involvement: it was curated by human workers who were asked to verify that each image contains a particular synset category (see [5] for a discussion of the collection process). While annotators were encouraged to select images regardless of occlusions, the number of objects and clutter in the scene, this process nevertheless produced a relatively clean dataset with fairly object-centric images. (3) Several of our experiments make use of DeiT-SIN [13], which is trained on stylised ImageNet [6] with labels. We don't believe that this supervision is critical, since in Fig. 4 of the main paper, we showed that ResNet50-MoCov2 [8] which does not use labels achieves similar performance (less than 1 mIoU difference on PASCAL-Context [12]). Moreover, we note that previous unsupervised methods to which we compare (e.g. [10, 3]) initialise their approach from supervised ImageNet training with the convention that *unsupervised* in this context denotes the fact that no pixel-level supervision is used. (4) By using CLIP [17], we also make use of a different kind of supervision, namely images paired with alt-text scraped from the web. Empirically, this data source has been shown to be extremely scalable and to enable generalisation to very large numbers of concepts [17, 11, 16]. Nevertheless, the creation of the original alt-text image descriptions (mostly) derives from a human source, and therefore provides a form of human supervision. In contrast to using ImageNet classification labels, this source of supervision *is* indispensable to ReCo.

We believe that the key factors to be considered when discussing the question of supervision are *scalability* and *generalisation*. We are typically not interested in unsupervised methods for their own sake, but rather because they offer the ability to cheaply scale up machine learning to larger training data sets that improve performance, and to build methods that go beyond the functionality afforded by labelled datasets (e.g. new classes, new tasks etc.). Subject to the caveats (e.g. human photographer bias) outlined above, we believe ReCo has the flexibility to scale up far beyond the experimental comparisons conducted in this work without requiring any changes to the underlying framework.

## A.2    Discussion of consent in used datasets

In this work, we work primarily with widely used Computer Vision benchmarks: ImageNet [5], PASCAL-Context [12], Cityscapes [4], COCO-Stuff [1], KITTI-STEP [21]. For these datasets, we do not conduct an independent investigation of consent beyond the considerations of the authors that released these datasets. For our final exploratory studies which make use of LAION-5B [18], we manually verified that no humans were present in the archives that were curated by ReCo.

### A.3 Discussion on whether data contains personally identifiable information or offensive content

We do not release any data as part of this work. By working with widely used Computer Vision benchmarks, we also restrict ourselves to imagery that is available in the public domain. We therefore believe that the risk that our work builds on harmful content or contributes to the leakage of personal information is low.

One exception to this is our use of the LAION-5B dataset for qualitative studies. We manually verified that no personally identifiable information or harmful content (as judged by the authors) was present in the archives curated by ReCo.

### A.4 Dataset licenses

Here we describe the terms/licenses of datasets used in our paper. For images in PASCAL-Context and COCO dataset, we comply with the Flickr Terms of Use and the Creative Commons Attribution 4.0 License for the COCO-Stuff annotations. For Cityscapes and ImageNet1K, we follow the terms stated on their official website[1] and the Attribution-NonCommercial-ShareAlike 3.0 Unported (CC BY-NC-SA 3.0) licence for KITTI-STEP.

## B Experiment details

Here we provide pseudocode for ReCo and details of experiments conducted in the main paper.

### B.1 Pseudocode for ReCo

In Alg. 1, we describe the pseudocode for the *core* of ReCo (to maintain readability, language-guided co-segmentation and context elimination are omitted since these follow a similar structure).

### B.2 Prompt engineering

To obtain the text embedding for a concept, we ensemble the textual features from 85 templates, *e.g.,* "a photo of the {concept}" and "there is a {concept} in the scene" following [22].

### B.3 Details of ablation study to assess CLIP retrieval performance

We observe that two of the ImageNet1K class labels are not unique—they occur twice with different meanings (*e.g.,* "crane" is used to represent both bird and machine), which makes retrieval inference and evaluation ambiguous. Therefore, we exclude those classes and use the remaining 996 categories for the experimental results reported in Fig. 4 (left) of the main paper.

### B.4 Hyperparameters for ReCo+ training

As described in Sec. 4.2 (in the main paper), we adopt DeepLabv3+[2] [2] with ResNet101 encoder [9] for ReCo+ and train the network with standard data augmentations such as random scaling and horizontal flip following [14, 19]. In detail, for geometric transformations, we use random scaling with a range of [0.5, 2.0], random crop with a crop size $320 \times 320$ pixels, and random horizontal flip with a probability of 0.5. For the photometric augmentations, we apply colour jittering[3] with 0.8, 0.8, 0.8, 0.2, and 0.8 for brightness, contrast, saturation, hue and probability parameters respectively. We also employ Gaussian blurring with a kernel size of 10% of $\min(H, W)$ where $\min(H, W)$ returns the length of the shorter side of an image.

---

[1] https://www.cityscapes-dataset.com/license and https://www.image-net.org/download.php for Cityscapes and ImageNet1K respectively.

[2] We use the code for DeepLabv3+ from https://github.com/VainF/DeepLabV3Plus-Pytorch.

[3] We use ColorJitter function in torchvision package [15].

**Algorithm 1** Pseudocode for the core of ReCo (using PyTorch-like syntax)

**Input.** a CLIP image encoder $\psi_\mathcal{I}$, a CLIP text encoder $\psi_\mathcal{T}$, an image encoder $\phi_\mathcal{I}$, an image collection $\mathcal{U}$, a concept $c$, the number of co-segmented images $k$

**Output.** a reference image embedding `ref_emb` and a prediction of the concept $c$ in a new image

```
# retrieve images
image_emb = l2_normalize(ψ_I(U), dim=1) # NxC
text_emb = l2_normalize(ψ_T(c), dim=0) # C
scores = mm(image_emb, text_emb) # N
indices = argmax(scores)[:k] # k
images = U[indices] # kx3xHxW

# co-segment
F = l2_normalize(φ_I(images), dim=1) # kxCxhxw
F_flat = F.permute(1,0,2,3).view(C,k*h*w) # Cxkhw
A = mm(F_flat.T, F_flat) # adjacency matrix, khwxkhw

grid = zeros((k*h*w, k))
start_col = 0 # start column index
for i in range(k):
    end_col = start_col + h*w # end column index
    grid[:,i] = max(A[:,start_col:start_col+end_col], dim=1)
    start_col = end_col
avg_grid = mean(grid, dim=1) # khw

seed_features = []
start_row = 0 # start row index
for i in range(k):
    end_row = start_row + h*w # end row index
    index_1d = argmax(avg_grid[start_row:end_row])
    start_row = end_row
    index_2d = [index_1d//w,index_1d%w]
    seed_features.append(F[i,:,index_2d[0],index_2d[1]])
seed_features = stack(seed_features, dim=0) # k×C
ref_emb = l2_normalize(seed_features.mean(dim=0), dim=0) # C

# inference
F_new = l2_normalize(φ_I(new_image), dim=0) # Cxhxw
prediction = sigmoid(mm(ref_emb, F_new)) # hxw
```
mm:matrix multiplication.

| tree | sky | building | road | person | mIoU |
|------|-----|----------|------|--------|------|
| ✗ | ✗ | ✗ | ✗ | ✗ | 5.7 |
| ✓ | ✗ | ✗ | ✗ | ✗ | 10.9 |
| ✓ | ✓ | ✗ | ✗ | ✗ | 12.0 |
| ✓ | ✓ | ✓ | ✗ | ✗ | 10.8 |
| ✓ | ✓ | ✓ | ✓ | ✗ | 11.4 |
| ✓ | ✓ | ✓ | ✓ | ✓ | **12.3** |

| parking →parking lot | vegetation →tree | mIoU |
|----------------------|------------------|------|
| ✗ | ✗ | 15.4 |
| ✗ | ✓ | 16.0 |
| ✓ | ✗ | 18.6 |
| ✓ | ✓ | **19.3** |

Table 1: **Effect of context category choices and reducing label ambiguity. Left:** We find that suppressing 5 frequently appearing categories brings performance gain on PASCAL-Context [12]. **Right:** We observe that specifying the meaning of a class more concretely helps segmentation performance of ReCo on Cityscapes [4]. For both cases, input images are resized and center-cropped to 320×320 following [7].

| Model | CLIP arch. | DenseCLIP arch. | mIoU |
|---|---|---|---|
| | ResNet50 | ResNet50 | 20.3 |
| ReCo (ours) | ViT-B/32 | ResNet50 | 20.3 |
| | ViT-L/14@336px | ResNet50x16 | 22.0 |

| # seed pixels | mIoU | Acc. |
|---|---|---|
| 1 | 22.0 | **65.4** |
| 5 | **22.2** | 64.9 |
| 10 | 22.1 | 62.9 |
| 50 | 21.7 | 57.5 |
| 100 | 20.4 | 51.0 |

Table 2: **Effect of architecture choices (left) and the number of seed pixels (right) on performance of ReCo on the Cityscapes validation set.**

## C   Additional ablation studies

### C.1   Choices of context categories

As described in Sec. 3.3 (main paper), we propose to suppress the commonly appearing categories, *e.g., sky*, which co-occur with other classes, *e.g., aeroplanes*. To achieve this, we manually pick 5 frequently appearing classes in PASCAL-Context dataset [12] and investigate the effect of different combinations of such categories. In Tab. 1 (left) we observe that suppressing the *tree* and *sky* categories yields a notable performance gain, while eliminating all five categories performs best. For this reason, we apply the context elimination strategy with these five categories in the main paper.

### C.2   Category name rephrasing to reduce ambiguity

We observe that it is important to specify a concept concretely to obtain retrieved images exhibiting similar visual appearance. For instance, in Cityscapes dataset, *parking* and *vegetation* can be rephrased to less ambiguous concepts *parking lot* and *tree* respectively based on their descriptions[4] in the paper accompanying the dataset [4]. As can be seen in Tab. 1 (right), ReCo gains benefits in performance on Cityscapes by replacing the category names with less abstract concepts. This sensitivity is a consequence of our co-segmentation algorithm, which locates pixels that share similar visual features across multiple images. Thus we use the rephrased label names throughout the experiments in the paper.

In addition to the limitations listed in the main paper, this dependence on concrete/specific concept names can be considered a limitation of our approach (albeit one that is readily mitigated). However, we believe it is a reasonable requirement for methods that operate in the zero-shot transfer setting. Unlike fine-tuning methods that learn to associate abstract text descriptions to visual concepts by seeing examples from the target distribution, ReCo relies entirely on an adequate text description to disambiguate the concept. Since many computer vision datasets have been constructed with training and testing splits with the assumption that methods would *make use of the training set*, we believe it is probable that category names were not designed to be uniquely descriptive (hence the use of "parking" as a category in Cityscapes, which could be either a verb or a noun). Indeed, there may have been little perceived need to construct unambiguous category names when examples from the training set implicitly provide disambiguation of the concept.

### C.3   Effect of architectural choices for CLIP and DenseCLIP

For the experiments in the main paper, we use ViT-L/14@336px and ResNet50x16 for image retrieval and DenseCLIP inference respectively. As these models are relatively heavier than commonly used architectures such as ResNet50 or ViT-B/32, we evaluate ReCo with lighter encoders on the Cityscapes validation split in Tab. 2 (left). Specifically we use either ResNet50 or ViT-B/32 for image retrieval and ResNet50 for DenseCLIP inference. As can be seen, adopting a lighter model slightly decreases the performance, but still outperforms previous state-of-the-art methods (*e.g.,* 16.3 mIoU for D&S [20]).

---

[4]"Horizontal surfaces that are intended for parking and separated from the road, either via elevation or via a different texture/material, but not separated merely by markings." for *parking* and "Trees, hedges, and all kinds of vertically growing vegetation." for *vegetation*.

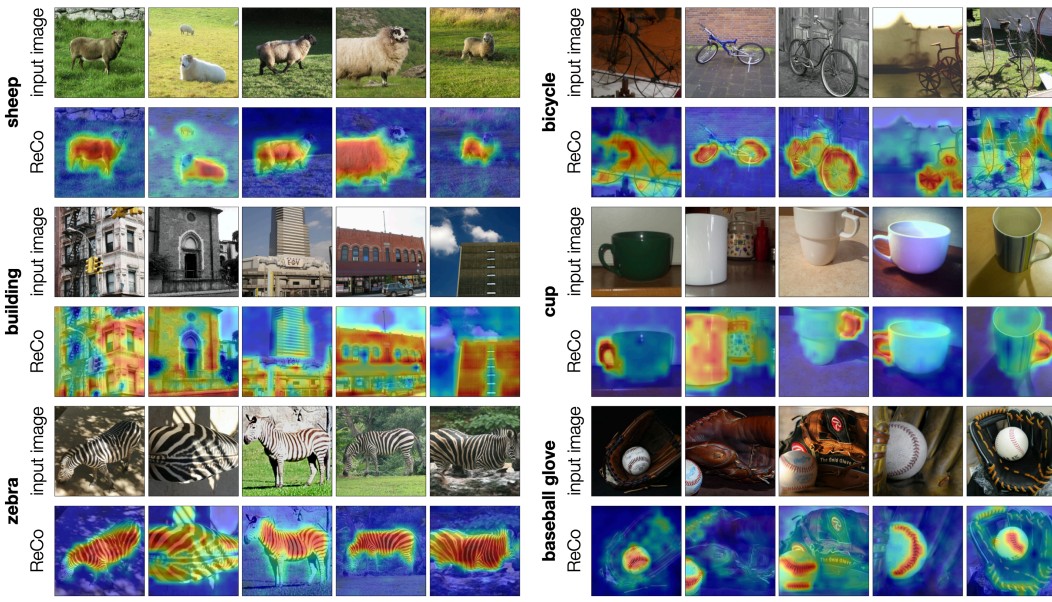

Figure 1: **Additional visualisations of our co-segmentation method used for ReCo.** For visualisation purpose, we show the top 5 images from 50 retrieved images in each archive for a category. **Left:** Successful cases. **Right:** Typical failure cases. Highlighted regions are shown in red. Best viewed in colour.

### C.4 Effect of the number of seed pixels

While we pick one seed pixel per image by default as decribed in Sec. 3, we investigate how the number of seed pixels selected for an image affects performance of ReCo. For this, we evaluate ReCo on Cityscapes with each reference image embedding computed by averaging 1, 5, 10, 50, and 100 seed pixel(s) for an image. As can be seen in Tab. 2 (right), using one or five seed pixel(s) per image shows the best performance in terms of pixel accuracy and mIoU and picking more pixels tends to hurt the performance.

## D Additional visualisations

### D.1 Co-segmentation with seed pixels

In Fig. 1, we visualise examples of the co-segmentation with seed pixels on ImageNet2012, which is used for an index dataset for ReCo in the main paper. On the left, we show successful cases where the co-segmentation highlights regions corresponding to a given concept (*i.e.,* sheep, building, and zebra). On the right, we display examples of two typical failure cases: partial segmentation (*i.e.,* bicycle and cup) and highlighting an object commonly co-occurring with a given concept (*i.e.,* baseball glove). For bicycles, their frames are less highlighted compared to the wheels. Similarly, the body parts of the cups are less likely to be emphasised than the handles. In case of baseball gloves, the co-segmentation locates a part of a baseball, which often appears with a baseball glove. We believe these failure cases are caused by the property of our co-segmentation algorithm, which focuses on regions with less variance in visual features (e.g., texture and shape) appearing in multiple images of an archive.

### D.2 Predictions of ReCo and ReCo+

In Fig. 2, we show more visualisation samples on the COCO-Stuff benchmark. Successful and failure cases are shown on the left and right, respectively. We note that ReCo tends to fail in predicting small objects, *e.g.,* people in the bus, and so does ReCo+ which is trained on the ReCo's predictions as pseudo-labels. We conjecture that this is related to the stride of the image encoder used for ReCo,

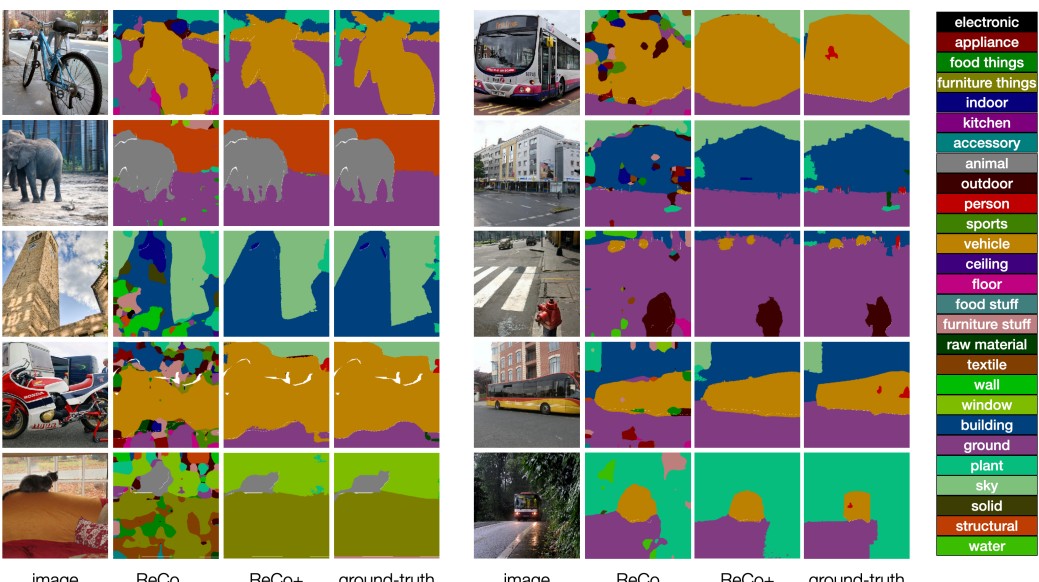

image    ReCo    ReCo+    ground-truth      image    ReCo    ReCo+    ground-truth

Figure 2: **Additional visualisations on COCO-Stuff. Left:** Successful cases. **Right:** Typical failure cases. White pixels denote ignored regions.

which is 16×16 for the case of DeiT-S/16-SIN [13]. It could therefore potentially be improved by using an encoder with a smaller stride at the cost of increased computational burden.