# OpenReview forum: "ReCo: Retrieve and Co-segment for Zero-shot Transfer"
_NeurIPS.cc/2022/Conference — NeurIPS 2022 Accept_

### Official Review · Reviewer_uo4A · 2022-07-10

**Rating:** 5
**Confidence:** 3
**Soundness:** 3 good
**Presentation:** 2 fair
**Contribution:** 3 good

**Summary:**

This paper utilizes the CLIP model for zero-shot transfer. At first, they leverage the CLIP to dynamically curate training sets from unlabelled images for arbitrary collections of concept names and leverage the robust correspondences offered by modern image representations to co-segment entities among the resulting collections. The synthetic segment collections are then employed to construct a segmentation model whose knowledge of concepts is inherited from the scalable pre-training process of CLIP. In this way, the proposed method could perform unsupervised segmentation approaches while inheriting the convenience of nameable predictions and zero-shot transfer.

**Questions:**

I only have one question about the fourth step in identifying seed pixels. Could the author clarify them?

**Ethics Review Area:**

["Responsible Research Practice (e.g., IRB, documentation, research ethics)"]

**Limitations:**

I am very curious about the potential of employing the proposed method in instance segmentation. And the whole training pipeline is a little complicated.

**Strengths And Weaknesses:**

strengths:

1, The paper is well written and easy to understand.
2, Leveraging the CLIP for unsupervised segmentation is interesting.
3, The proposed training pipeline is reasonable.
3, The experiments are sufficient to show the effectiveness of the proposed method.

weaknesses:

1, The whole training pipeline seems a little complex. For example, the proposed method should utilize CLIP to filter some candidate images from numerous unlabeled data. And the identification of seed pixels includes four steps
2, Also, the adjacency matrix A is the computation cost and is sensitive to k in the first step of the identification of seed pixels.

---

> ### Author Response · Authors · 2022-08-02
> **Response to Reviewer uo4A**
>
> We thank Reviewer uo4A for their valued comments, and finding our paper well-written and effective. Below we address the raised concerns.
>
> ### Conceptually complicated co-segmentation process
> Thank you for pointing this out. Indeed, CLIP is used for collecting unlabelled images to form an archive for a given concept, which is then used for the following co-segmentation through the four steps.
>
> While it is possible to simplify the co-segmentation process by replacing the seed identification with a clustering algorithm as we show in our response to **Reviewer Sw1r**, we think the proposed seed identification process is worth the conceptual complexity considering its superior performance compared to simple alternatives by around 5 mIoU.
>
>
> ### Is computing an adjacency matrix computationally costly?
> In our case, the computational cost required for computing an adjacency matrix is negligible compared to the whole inference. This is due to the small dimensions of features extracted from a visual encoder used in the paper (i.e., DeiT-S/16). For example, when feeding a 224x224 image as in our paper, its output features have a shape of 384x14x14. To construct an adjacency matrix across features from 50 images of 224x224, this results in 0.72G Multiply–accumulate operation (GMACs), which accounts for only 0.34% of the whole co-segmentation process including the feature extraction step (approximately 214.22 GMACs).
> To provide how the compute cost of constructing an adjacency matrix changes over an archive size, we experimented 5 different archive sizes from 10 to 50. For reference, we also report performance of ReCo on PASCAL-Context with each archive size:
>
> | archive size | GMACs | % (compared to whole inference) | mIoU |
> |:---:|:---:|:---:|:---:|
> |10|0.14| 0.34 | 25.8 |
> |20|0.29| 0.34| 26.5 |
> |30|0.43| 0.34| 26. 1|
> |40|0.58| 0.34| 27.3 |
> |50|0.72| 0.34| 27.2 |
>
> As can be seen, the adjacency matrix is not computationally intensive in our case due to the small dimensionality of output features from DeiT-S/16.
>
>
> ### How does the fourth step in the co-segmentation work?
> The fourth step is simply to obtain a spatial index of a seed pixel by applying an _argmax_ operator.
>
> To recall, given $k$ images from an archive, our aim is to pick a seed pixel for each image which shows _highest average maximum similarity_ across the images.
> For this, the first step is to construct an adjacency matrix between all features.
> This yields a matrix of shape $khw$ x $khw$, where $h$ and $w$ denote height and width of the dense features.
> The second step is to take the maximum similarity value between a pixel in an image and another pixel a different image. For this, we apply a _maxpooling_ operator with a kernel size of $khw$ x $hw$ and a stride of 1 x $hw$. This operation produces a matrix of shape $khw$ x $k$. The (i, j)th entry of this matrix (0 $\leq$ i $\leq khw-1$, 0 $\leq$ j $\leq k-1$) represents the maximum similarity value between the ith pixel and another pixel in jth image. The third step is to take average of columns to produce a $khw$ x 1 matrix of which element denotes an average maximum similarity across the $k$ images. As there are $hw$ pixels per image, we pick the one with highest average maximum similarity by applying an _argmax_ operator to each set of $hw$ pixels for an image, which is the fourth step. As a result, we obtain $k$ spatial indices, each of which denotes a location of a seed pixel for an image.
>
> ### Potential extension of our pipeline to instance segmentation
> We also think it would be very interesting to see how it unfolds for instance segmentation and hope to extend our framework to instance segmentation for our future work.

---

### Official Review · Reviewer_VKMh · 2022-07-11

**Rating:** 6
**Confidence:** 3
**Soundness:** 3 good
**Presentation:** 3 good
**Contribution:** 2 fair

**Summary:**

This paper proposed a method for zero-shot transfer in semantic segmentation. To solve this problem, it first performs a image-text retrieval by CLIP to get image archive. Then it use a pre-trained encoder to perform co-segmentation. During inference, it combines the results from reference image embedding and Dense CLIP to get the final segmentation results.

**Questions:**

see weakness

**Limitations:**

Yes.

**Strengths And Weaknesses:**

Strengths:
* The proposed pipeline which combines retrieval and co-segmentation is novel.
* It outperforms the compared methods significantly.

Weakness:
* The method is complicated and requires two encoders during inference, which slow down the speed. I want to know the comparisons in FPS when compared to other methods.
* Missing important citations: there are some concurrent works for open vocabulary semantic segmentation[1, 2], which are not cited. It would be better if related discussions are included.

[1] A simple baseline for zero-shot semantic segmentation with pre-trained vision-language model. Arxiv.

[2] Decoupling Zero-Shot Semantic Segmentation. CVPR 2022.

---

> ### Author Response · Authors · 2022-08-02
> **Response to Reviewer VKMh**
>
> We thank Reviewer VKMh for their valued feedback and for acknowledging the novelty of our work.
> In the following, we respond to their concerns and question.
>
> ### Does using two encoders slow down ReCo inference?
> Thank you for proposing this experiment. We measured the speed for a single forward pass of each model on the 500 images of the Cityscapes validation split with an RTX 2080 Ti graphic card. In detail, we resized and center-cropped images to 320$\times$320 following the inference setting in STEGO [2]. As the previous methods do not involve post-processing, we exclude the CRF post-processing for DenseCLIP [3] and ReCo.
>
> |method|encoder arch.|decoder arch.|FPS|mIoU|
> |:---|:---|:---:|:---:|:---:|
> |MDC$^\dagger$| RN18| FPN |67.0| 7.0 |
> |PiCIE$^\dagger$| RN18 | FPN | 75.7 | 9.7|
> |DenseCLIP| RN50 | - | 58.2 | 10.6|
> |DenseCLIP| RN50x16 | - | 34.6 | 9.7|
> |ReCo| RN50 & DeiT-S/16 | - | 35.6 | 17.6|
> |ReCo| RN50x16 & DeiT-S/16 | - | 25.1 | 18.5|
>
> $^\dagger$Numbers for mIoU are reported in D\&S [4].
>
> Indeed, as shown in the table, ReCo has a lower FPS compared to other models but not slow in terms of an absolute speed (35.6 FPS for ReCo with light encoders). While we believe this is not a big drawback of our approach, we plan to streamline the inference process with a single network for our future work. We added this point as a potential limitation of our approach in our revision.
>
> ### Missing citations
> Thank you for letting us know the missing citations. We added the citations of the papers and discussions in the modified version.

---

### Official Review · Reviewer_Gih1 · 2022-07-11

**Rating:** 6
**Confidence:** 5
**Soundness:** 4 excellent
**Presentation:** 3 good
**Contribution:** 3 good

**Summary:**

This paper addresses the task of zero-shot segmentation in images by leveraging powerful large-scale pretrained vision-and-language models such as CLIP. Interestingly, the proposed approach does not require costly and time-consuming pixel-wise annotations for training. Instead, it uses CLIP to select groups of relevant images that correspond to the natural language queries, based on nearest neighbors. Next, it uses pretrained visual encoders to identify seed pixels in the images that have strong support across the entire group of relevant images. These seed pixels are used to compute a reference feature for each language query to produce a segmentation attention map for each new query image, which is further refined by another segmentation mask that is computed by CLIP.

**Questions:**

1) During inference time, are you always computing the reference feature for a given image and query concept on the fly, or do you store a dictionary of precomputed reference features for a selected list of concepts?

**Limitations:**

Yes, the authors have addressed the limitations.

**Strengths And Weaknesses:**

Strengths:

1) The paper is largely well-written and easy to follow. In particular, the mathematical definitions that are provided are very helpful for understanding the proposed approach.

2) The proposed approach is theoretically sound and intuitive. While it is not entirely original due to the existence of approaches including DenseCLIP, the idea of discovering common spatial regions that occur in images containing the same concept is very interesting. More importantly, it leverages the large-scale pretrained CLIP model to retrieve related images for a language query. This allows the proposed approach to be trained on any unlabeled image sets.

3) The task of image segmentation often requires fine-grained pixel-wise annotations which is an especially costly process. Being able to leverage powerful and large-scale pretrained models to circumvent this process is especially significant. Coupled with the empirical evidence that it outperforms state-of-the-art approaches, this can be an important area of research, given the availability of increasingly larger multimodal datasets such as LAION-5B.

Weaknesses:

It would be helpful to see some qualitative visualizations of co-segmentation with seed pixels. Given that these seed pixels are used to compute a reference embedding for new query images and concepts during inference time, it seems to be a very important component of the proposed approach. It may help a reader to determine if the regions selected by the seed pixels are consistent across most images that contain a concept.

---

> ### Author Response · Authors · 2022-08-02
> **Response to Reviewer Gih1**
>
> We thank reviewer Gih1 for their valued feedback and for finding our approach theoretically sound and a potentially important area of research. Below we address their concerns and questions.
>
> ### Qualitative visualizations of co-segmentation with seed pixels
> Thank you for suggesting this - we added visualisation samples for both successful and failure cases of the proposed co-segmentation algorithm in appendix.
>
> ### Does ReCo store a dictionary of precomputed reference features for a selected list of concepts?
> Yes, we store a dictionary of reference features so that inference can be faster. To clarify this in the paper, we added the following sentence: "For each benchmark, we pre-compute reference image embeddings for a list of categories for the benchmark and store the embeddings to form a classifier." in Sec. 4.2 of our revision.

---

> > ### Comment · Reviewer_Gih1 · 2022-08-09
> > **Reply to authors**
> >
> > Thank you very much for adding the visualization samples as well as the detailed responses to all of our questions. I agree with most of the strengths that are raised by other reviewers and the authors have also answered my question satisfactorily. I will retain my initial rating of weak accept.
> >
> > Please incorporate the comments raised by other reviewers as they are very helpful towards improving the quality of the paper.

---

### Official Review · Reviewer_Sw1r · 2022-07-13

**Rating:** 4
**Confidence:** 3
**Soundness:** 2 fair
**Presentation:** 2 fair
**Contribution:** 2 fair

**Summary:**

This paper proposes a retrieve and co-segment approach that leverages a pretrained image-text model (e.g. CLIP) for unsupervised semantic segmentation. The results on existing benchmarks are good compared to other unsupervised segmentation approaches.



**Questions:**

See weakness.

**Limitations:**

Yes.

**Strengths And Weaknesses:**

Strength
* Retrieve and co-segment is an intuitive and reasonable approach for unsupervised segmentation.
* Using image-text models e.g. CLIP for retrieval makes sense and is effective.
* Adaptation to target distribution by training on pseudo labels is reasonable and effective.
* Performance of ReCo+ seems better than existing unsupervised segmentation approaches at system level.

Weakness
* Compared with the unsupervised segmentation approaches, I think ReCo has a clear advantage by using CLIP which makes the approaches not directly comparable. CLIP has seen lots of image-text pairs and acquired reasonable pixel localization ability, while existing approaches such as PiCIE have no access to this kind of knowledge. In addition, if I understand correctly, ReCo has access to the category names of the target dataset while existing approaches do not.
* ReCo uses ViT-L/14 for retrieval, which is larger and stronger than the models used by existing works (e.g. ResNet18 of PiCIE). How does the performance of ReCo compare to existing unsupervised segmentation methods if we use smaller CLIP models (e.g. R50 or ViT-B/32) for retrieval and inference?
* How does the performance change if you pick more than one seed pixel per image?
* The steps to identify seed pixels (L158-168) seem highly heuristics-based. Alternatively, would clustering approaches work there?

---

> ### Author Response · Authors · 2022-08-02
> **Response to Reviewer Sw1r**
>
> We thank Reviewer Sw1r for their valued comments. We address their comments below and include additional experiments in our revision.
>
> ### Is ReCo directly comparable to previous unsupervised semantic segmentation methods?
> We compared to unsupervised baselines since we believe that they were the most appropriate baselines among existing work (similarly to ReCo, they make no use of pixel-level human annotation).
> However, the use of image-text pairs during pre-training (which is central to our approach) does represent a different kind of supervision (this topic is discussed in some detail in the supplementary in Sec. A.2). To clarify the distinction, we
> will update the submission to name our task setting "unsupervised semantic segmentation _with language-image pre-training_" at the beginning of Sec. 3.
> ReCo does indeed have access to the category names - it is this access that enables it to perform named predictions, a key benefit of the approach. Please note that conventional unsupervised methods are given the number of categories and pixel-level ground-truth masks to determine the name of each prediction by a matching algorithm during inference.
> We will elaborate the differences from USS in our final version given an additional page upon acceptance (9-page limit applies during author-reviewer discussion period and we have no space to describe the differences in detail).
>
> ### How does ReCo perform with smaller CLIP models?
> When we use ResNet50 or ViT-B/32 for retrieval and ResNet50 for DenseCLIP inference, ReCo shows following results on Cityscapes:
>
> |CLIP|DenseCLIP|mIoU|
> |:---|:---|:---:|
> |ResNet50|ResNet50| 20.3 |
> |ViT-B/32|ResNet50 | 20.3 |
> |ViT-L/14@336px | ResNet50x16 | 22.0 |
>
> It is worth noting that the numbers shown in the table are different from Tab. 2 in the submitted paper. We re-run ReCo at original resolution for a fair comparison.
>
> When compared to the existing methods, ReCo still shows a favourable performance with lighter architectures. We added this experiment in the appendix.
>
> ### Is it optimal to pick 1 seed pixel per image?
> The following table shows performance of ReCo on Cityscapes when we pick $N$ seed pixels per image with $N$ = {1, 5, 10, 50, 100}:
>
> |$N$|mIoU|Acc.|
> |---:|:---:|:---:|
> |1|22.0|**65.4**|
> |5|**22.2**|64.9|
> |10|22.1|62.9|
> |50|21.7|57.5|
> |100|20.4|51.0|
>
> Picking 1 or 5 seed pixel(s) shows the best performance compared to the cases with more seed pixels. When comparing the two cases, picking 5 seed pixels performs slightly better than 1 pixel w.r.t. mIoU whereas picking 1 pixel is better w.r.t. pixel acc. We added this experiment in the appendix.
>
> ### Can we replace the seed identification process with a less heuristic way, e.g., by utilising a clustering algorithm?
>
> We experiment to replace the proposed seed identification method with two different methods involving a clustering algorithm. For each approach, we test $k$-means and spectral clustering with different cluster sizes ($k$). All methods are evaluated on Cityscapes in terms of mIoU.
>
> For the first method, we simply apply a clustering algorithm across the whole features from all images in an archive with an assumption that one of the resulting groups represents regions for the category. To decide which cluster represents the class, we select a cluster with the lowest average pair-wise L2 distance between its features, assuming an intra-class variation within a cluster for a concept is smaller than that within other clusters. Then, we average the features belonging to the selected cluster for a reference image embedding:
>
> |method|$k$=$2$|$k$=$3$|$k$=$4$|$k$=$5$|$k$=$6$|$k$=$7$|$k$=$8$|
> |---:|:---:|:---:|:---:|:---:|:---:|:---:|:---:|
> |$k$-means|  -$^*$ |10.5|11.3|10.9|12.0|11.1|11.3|
> |spectral | -$^*$ | 10.6 | 12.3 | 11.1 | 10.6 | 11.3 | 11.1|
>
> $^*$Not implemented due to our limited computational resources.
>
> While the performance is comparable to PiCIE (9.7), the proposed method (22.0) shows better performance than applying a clustering algorithm.
>
> As a second method, given $k$ clusters from an application of clustering to features of an image, we use CLIP to decide which cluster represents a category region in an image. We feed CLIP a masked input whose non-highlighted pixels are zeroed and output a probability of the masked image belonging to the class. By comparing $k$ probabilities of an image masked by one of $k$ clusters for the image, we pick a cluster which leads to the highest class probability.
> We gather features of a selected cluster for each image and set the average of the features as a reference image embedding for the concept:
>
> |method|$k$=$2$|$k$=$3$|$k$=$4$|$k$=$5$|$k$=$6$|$k$=$7$|$k$=$8$|
> |---:|:---:|:---:|:---:|:---:|:---:|:---:|:---:|
> |k-means| 15.7 | 16.5 | 17.3 | 16.5 | 17.5 | 17.2 | 17.6 |
> |spectral | 15.9 | 16.0 | 16.7 | 17.5 | 17.5 | 16.7 | 16.8|
>
> Overall performance is comparable to the state-of-the-art (i.e. 16.3 for D\&S). However, it is not as good as the proposed method (22.0).

---

> > ### Comment · Reviewer_Sw1r · 2022-08-09
> > **Comments well addressed - upgrade score.**
> >
> > The authors address my questions well, and I'll upgrade my score to 6. I appreciate the experiments on smaller CLIP models, # of seeds, and applying clustering for seed selection. Some additional comments:
> >
> > - It's nice to see performance holds up well for the smaller CLIP models. I'm surprised the mIoU gap is only 2 between R50x16 and R50. This is good and bad because it implies 1) you can use small model to get good performance, and 2) larger models do not help you too much. Any guess why larger models are not much better?
> > - The experiment on clustering is interesting and surprising that it does not outperform the heuristics-based approach. Perhaps there's something in the heuristics (e.g. max operation) that should be incorporated somehow. We can defer this to future studies.
> > - Apart from renaming the task setting at the beginning of Sec. 3, it would be helpful to incorporate some of the response and A.2 to related parts of the manuscript (e.g. related work, intro, experiment) to clarify the comparison with existing unsupervised semantic segmentation works.

---

> > > ### Author Response · Authors · 2022-08-09
> > > **Response to post-comment of Reviewer Sw1r**
> > >
> > > We appreciate the post-comments of Reviewer Sw1r.
> > >
> > > ### Why don't larger models perform much better than smaller ones?
> > > Considering that larger CLIP models show much higher accuracy than lighter ones on image retrieval as shown in Fig. 1 (left), we conjecture that the reason why the larger models perform only slightly better on segmentation is due to the discrepancy between the upstream and downstream tasks. In other words, as the CLIP training does not involve segmentation but finding a correct pair of a text and an image, there seems to be an only weak connection between classification and segmentation performance of a CLIP model.
> > >
> > >
> > > ### Why don't the clustering methods outperform the heuristic-based approach?
> > > We think this is due to false positive pixels selected as a part of a category object, when applying a clustering algorithm. Specifically, the tested clustering methods above either clusters the whole features of images of an archive or features within an image. However, this may not be robust enough to properly handle objects or stuff that co-occur with a given category. For example, *road* pixels can frequently appear *car* pixels. As visual appearance within *road* regions can be less likely to vary more than appearance within *car* regions, a clustering-based method may primarily group *road* regions and wrongly pick the cluster for *road* instead of *car* from the *car* images.
> > >
> > > On the other hand, the proposed max operation seems to be less prone to select a wrong pixel as it does not need to consider the variation in appearance of a category within an image, but needs to find a combination of a single pixel for each image that shows a highest average maximum similarity. It is worth noting that this does not mean the max operation is entirely free from co-occurring objects/stuff and that is why incorporating the context-elimination process brings a notable performance gain (by 4.2 mIoU on PASCAL-Context as in Tab. 1).
> > >
> > > ### Suggestion for clarifying the task in the revision
> > > We agree it would be helpful for readers to be more clearly aware of the differences between the conventional unsupervised semantic segmentation and the considered task in the paper. We will further clarify this in introduction, related work, and experiments as well as method part (i.e., Sec. 3).
> > >
> > > We thank Reviewer Sw1r again for their insightful comments.

---

### Official Review · Reviewer_1QtT · 2022-07-14

**Rating:** 4
**Confidence:** 5
**Soundness:** 3 good
**Presentation:** 3 good
**Contribution:** 3 good

**Summary:**

This paper leverages CLIP for Zero-shot segmentation, which is a very hot topic currently. The authors proposed a CLIP-retrieval-based way to build gallery candidates for the semantics segmentation class, and then use dense-clip to generate reference image embedding. Then they proposed  several way to boost the performances, including language-guided co-segment and context elimination to remove the bias of background. Experiments show the proposed method achieves state-of-the-art performance.

**Questions:**

See weakness

**Limitations:**

Yes

**Strengths And Weaknesses:**

Strength:
- Outperforming State-of-the-art performance.


Weaknesses:
- 1. The paper is not well written. The authors make it hard to understand even for some very clear concepts.
- 2. Novelty is somehow limited. Technical contribution is not enough. It is just like to find a prototype for a class and then use it for normal clip inference. Although there are some modifications, such as context elimination, however, these are more like tricks which does not have technical depth.
- 3. Why the used numbers in Table 2 are different for DenseCLIP [92] (Table 1 and Table 3 in original DenseCLIP [92])?

---

> ### Author Response · Authors · 2022-08-02
> **Response to Reviewer 1QtT**
>
> We thank Reviewer 1QtT for their valued comments.
>
> ### Hard to understand even for some very clear concepts
> Are there are some specific concepts that the reviewer considers to be poorly communicated? If so, we are happy to propose revisions in order to address the reviewer's concerns.
>
> ### Limited technical novelty
> The heart of our contribution is a new framework that removes the requirement for pixel-level ground-truth labels typically required for named predictions in previous unsupervised methods. We leverage the unique curation abilities of modern vision and language retrieval models to reduce the hard task of semantic segmentation to the more manageable task of cosegmenting a curated archive. While novelty is inherently subjective, we believe that this represents a novel and significant contribution.  The remaining technical contributions are simple, and we do not view that as a negative. However, they do play a key role in enabling ReCo to work effectively. In Table 1, we ablate the effect of each component, illustrating the significant performance gap of 21.5 mIoU between a variant of ReCo without using any of the proposed components and another with all components.
>
> ### Why are the numbers for DenseCLIP in Table 2 different from the original paper?
> There are two key differences.
>
> As noted in the work in Section 4, we follow PiCIE [1], MDC [1] and STEGO [2] and evaluate on 27 mid-level categories, while the DenseCLIP [3] authors evaluate on 171 low-level categories.
> Second, we follow the evaluation protocol of STEGO, resizing and center-cropping input images to 320x320 pixels, whereas DenseCLIP evaluates at the original resolution of images.
> To control for these differences, we compute DenseCLIP performance ourselves and report it directly for comparison.

---

> > ### Comment · Reviewer_1QtT · 2022-08-09
> > **Thanks for the author response**
> >
> > Thanks for the response. I also checked the response to reviewer Sw1r, which is very helpful.
> >
> > For the concerns,
> >
> > 1) Presentation issue. Generally I think there are some simpler ways to present the idea, such as using some commonly used concepts such as prototypes for seed pixels, or background bias removal for context elimination. The main purpose of presentation is to make idea simple and clear, rather than complicated. I think it is better to add some illustrations in the current draft to call back some concepts that people are all familiar.
> >
> > 2) Although the response tried to defense the novelty issue, I am still not fully convinced. It is hard to say there are some general takeaways from this paper. Most of the improvements are from some tricks (e.g., context elimination, which is to suppress background probabilities). So my biggest concern is that this paper does not have a very clear technical contribution/inspiration idea. It is more like a bundle of tricks.
> >
> > 3) Thanks for the explanation of experiments. I have another question, what if remove Eq. (3)? It seems P^c_{new} is already good enough from Figure 2.
> >
> >
> > I would like to hear more about the contribution/novelty part.

---

### Author Response · Authors · 2022-08-02
**Thanks for the reviews and summary of changes in an updated paper.**

We thank all of the reviewers for their time and insightful comments.

The reviewers appreciated our work in terms of
(i) strong performance (
Reviewer 1QtT,
Reviewer Sw1r,
Reviewer VKMh
);
(ii) intuitive and interesting idea (
Reviewer Sw1r,
Reviewer Gih1,
Reviewer uo4A
);
(iii) good presentation (
Reviewer Gih1,
Reviewer uo4A
).

As the concerns and questions raised by each reviewer are mostly non-overlapping, we leave our response to each review.


We summarise main changes made in our updated paper (highlighted in red in the updated paper) in the following:

(i) Added an experiment with lighter architectures for ReCo in the appendix.

(ii) Added an experiment with multiple seed pixels per image for ReCo in the appendix.

(iii) Added important missing citations in Sec. 2.

(iv) Added visualisations of the proposed co-segmentation algorithm in the appendix.



*References*

[1] Picie: Unsupervised semantic segmentation using invariance and equivariance in clustering. In CVPR, 2021

[2] Unsupervised semantic segmentation by distilling feature correspondences. In ICLR, 2022

[3] Denseclip: Extract free dense labels from clip. arXiv:2112.01071, 2021

[4] Drive\&segment: Unsupervised semantic segmentation of urban scenes via cross-modal distillation. arXiv:2203.11160, 2022.

---

### Meta-Review · Area_Chair_QHjE · 2022-08-29

**Recommendation:** Accept
**Confidence:** Certain

**Metareview:**

After author response and the discussion the paper received 1x borderline reject, 1x borderline accept, 3x weak accept [note that one reviewer mentioned the score increase only in the discussion].

The main strength are:
- Overall novel framework for zero-shot segmentation
- Strong performance
- The authors revised the paper and addressed many/most of the reviewer's concerns/suggestions in the author response.

I recommend acceptance, with the expectation
* the authors provide the additional revisions as promised
* If possible address the comment of reviewer 1QtT "what if remove Eq. (3)? It seems P^c_{new} is already good enough from Figure 2."


**Award:**

No

---

### Decision · Program_Chairs · 2022-09-14

Accept